# Simulation of Groundwater Flow in an Aquiclude for Designing a Drainage System during Urban Construction: A Case Study in Madrid, Spain

**Joaquín Sanz de Ojeda [1,\*], Eugenio Sanz [2], Francisco Javier Elorza [1] , Cesar Sanz Riaguas [3] and Manuel de Pazos Liaño [4]**

1. Escuela Técnica Superior de Ingenieros de Minas y Energía, Universidad Politécnica de Madrid, c de Ríos Rosas, 21, 28003 Madrid, Spain; franciscojavier.elorza@upm.es
2. Laboratorio de Geología, Departamento de Ingeniería y Morfología del Terreno, Universidad Politécnica de Madrid, c Profesor Aranguren s/n, 28040 Madrid, Spain; eugenio.sanz@upm.es
3. Desarrollos Logísticos y Fomento de Suelo S.L. (DELFOS), c. de Narvaez, 15, 28009 Madrid, Spain; csanz@grupodelfos.net
4. Ayuntamiento de Madrid, Área de Gobierno de Obras y Equipamientos, Dirección General del Espacio Público, Obras e Infraestructuras, Subdirección General de Control de la Urbanización, c Montalbán, 1, 28014 Madrid, Spain; pazoslm@madrid.es
\* Correspondence: joaquin.sanzdeojeda@alumnos.upm.es

**Abstract:** A detailed hydrogeological study was carried out due to the recent occurrence of unexpected problems associated with the flooding of the water table during excavations in the area of a major urbanization work in Madrid. The numerous exploratory drilling excavations carried out allowed for the development of a conceptual model of the complex hydrogeological functioning in clay formations in an urban area. The clays have very little natural recharge, and the underground flow is highly conditioned by the topography and a fold-fault. Modelling with MODDFLOW confirmed and quantified this conceptual model and also allowed for the design of an efficient network of 1.5 km-long drainage trenches. The design of this drainage network was influenced by the difficult balance that must be respected in order not to contaminate the water with sulphates from the nearby gypsum substrate. This is to guarantee the quantitative and qualitative sustainability of the groundwater. The follow-up and monitoring of the water tables and the quality of the groundwater for more than a year after the excavation of the drainage trenches guaranteed the results of the research.

**Keywords:** Spain; urban groundwater; conceptual model; numerical modelling; groundwater



## 1. Introduction and Objectives

The growing trend for city managers to move towards sustainability by, for exampling, implementing sustainable urban drainage systems (SUDS), is helping to promote the recognition of groundwater as a key component in the urban planning process. In the coming years, the cumulative impact of subsoil structures on urban groundwater will increase. Future urban planning will have to combine the vertical development of a city and the sustainable development of groundwater. This challenge requires risk management, which is inherent to the evolution of urban groundwater flow in terms of quantity and quality.

In order to properly plan the development of urbanizations and to ensure the sustainability of groundwater, it is necessary to consider hydrogeological interactions from the earliest stages of construction. Excavation and foundation works, although temporary, are becoming increasingly deeper due to urban densification (see works under [1–6]). The lack of experience and understanding of the interactions between the different kinds of infrastructure (for example, levelling, permeable pavements, sewage systems, deep foundations, tunnels, and underground car parks) with urban aquifers can generate risks and problems,

modifying the flow and natural quality [7,8], soil compaction [9–11], corrosion of foundations, flooding, etc. [12]. Urbanization also disturbs the balance of the hydrological cycle, especially due to the alteration of runoff, which results in a modification of its recharge [13], and the implementation of drainage networks can reduce water levels [14].

As is well known, many construction engineering projects involve excavation below the water table, and it is common practice and sometimes essential to use drainage and groundwater control techniques. Groundwater drainage, as well as surface water drainage, is very important and, if not properly controlled, can lead to changes in construction methods and progress. Basically, there are two methods for controlling groundwater in construction areas: lowering the water table by drainage or pumping and excluding water flow inlets by making waterproof walls [15].

Walls may be indicated for the drainage of specific works (building foundations, garages, basements, subways, etc.), but for generalized interactions with the water table in large and extensive excavations, which are underway in Madrid, drainage by means of trenches is normally used.

Understanding groundwater flow before construction, or at least in the early stages of construction, is the first step to improve its control and management, although it will be necessary to know the possible disturbance in the natural flow system and modification of the quality of the groundwater induced by an excavation afterwards. In this sense, the use of numerical simulation methods is a capable tool for the representation and precise quantification of flow alterations. It allows for consideration of the 3D geometry of both the geology and the drainage systems required during excavation. In our case, it has been a particularly relevant tool for evaluating the efficiency of a network of drainage trenches in the framework of a complex hydrogeological context, involving the presence of two superimposed aquicludes with a fold and a fault that condition groundwater flow.

The objectives of this work are:

- To establish the conceptual hydrogeological model of an important urbanization work in Madrid (Los Ahijones, Vicálvaro district) and its surroundings; and
- To simulate the underground flow of the Miocene aquiclude, where a part of the urbanization work and its surrounding environment is located, in order to validate and improve the knowledge of the hydrogeological conceptual model. This simulation will serve to quantify the flow in a natural and influenced regime, evaluating the best solution to successfully carry out the construction of the urbanization work and the infrastructure it entails, as well as monitor it in the future.

## 2. Methodology

The chronological steps taken in this work were as follows:

- Hydrogeological knowledge was obtained prior to the start of the urbanization work, which comes mainly from numerous geotechnical and environmental studies of this work and other urbanization works in the surrounding area (for example, information on the water table from 19 boreholes). This, among other things, made it possible to draw up an equipotential lines map in 2006, before work on the development began.
- A detailed geological map on a scale of 1:2000 in the construction phase of the urbanization work was developed in 2019, complemented by soil pits and piezometers.
- A mesh of 90 soil pits, excavated to the level of the sewerage network (the deepest), was regularly distributed over the study area in order to ascertain the situation of the water table and the detailed stratigraphy in the surface area. They were also used to carry out some permeability tests. Thirteen of them were placed on either side of a drainage trench, up to 50 m away, as observation piezometers in order to spatially characterize the influence of the ditch on the depression of the water table (and therefore its efficiency) and the evolution of this decrease over time.
- 16 boreholes were drilled, which were between 10 and 30 m deep and arranged in a regularly distributed mesh within the urbanization work. This remains as a piezometric control network for the future. In addition to obtaining information from

the lithological column, 18 Lefranc-type permeability tests were carried out, the water table levels were measured regularly, and water samples were taken for chemical and isotopic analysis.

This rather large and evenly distributed set of piezometers and soil pits was complemented with information from boreholes and soil pits from the 2006 geotechnical projects and the inventory of water points in "Los Ahijones" creek basin. This allowed for: A. The geometrization of the various geological layers that could, in principle, enter into the hydrogeological game, defining their folds and faults that could condition the underground flow, apart from the topography. B. The estimation of the hydrogeological parameters of the different hydro-stratigraphic units. The estimation of permeability tests is not only carried out with the Lefranc tests, but also with pumping tests in alluvial deposits (one test) and in trial pits. The test in the alluvial deposit has served to confirm the orders of magnitude of the values obtained by Lefranc. The tests in the trial pits were the same but were few in number. In the gypsums, no tests, other than those by Lefranc, were carried out. C. The drawing up of an equipotential lines map in August 2019. All this was conducted in order to establish the conceptual model of hydrogeological operation.

- In order to improve the conceptual hydrogeological model, it was important to know the hydrogeochemistry of the area, such as the characterization of hydrochemical facies, origin of recharge, etc. For this reason, 12 water samples were taken in the field at the most representative points using piezometers, soil pits, drainage trenches, springs, etc., with in situ measurements of the conductivity, temperature, and pH of the water. For the piezometers, a sample-taker was used to determine the chemical stratification of the groundwater. Samples were also taken from 7 representative points of water for analysis of the environmental isotopes (Deuterium and Oxygen 18) in order to determine the origin of the natural recharge.
- A hydrogeological conceptual model was used to simulate the underground flow situation with MODFLOW [16], before work began in 2006. At this time, the northern part of the basin (Cañaveral urbanization) was not urbanized. This modelling allowed us to verify and perfect the previous conceptual hydrogeological model. Then, we simulated the situation corresponding to April-September 2019, when The Cañaveral had already been urbanized, and once the excavations of a part of the "Los Ahijones" urbanization work and a drainage ditch of one kilometer in length had been carried out. The rest of the circumstances did not change between the two situations. Once the mathematical model was calibrated, the effect on the phreatic of the drainage system proposed in a transitional regime and the possible decrease in the future recharge of the basin by waterproofing due to the construction of other urbanizations were simulated.

## 3. Description of the Site and the Problem

The study focuses on a part of the "Los Ahijones" urbanization work, which is located in the Vicálvaro district of Madrid (Figure 1). It has an area of 550 ha. The area under study comprises the surface area (6.93 km$^2$) of the hydrological basin of the Arroyo of the same name.

This stream of "Los Ahijones" intercepts small streams on its right bank, which provide it with runoff water in episodes of rain. These are two small valleys (also called Paleobasins), which are almost always dry:

- Thalweg of "Cañaveral", which we have named after this urban area. It is always dry except when it receives contributions from drainage infrastructure in Radial 3 (R-3 in Figure 1).
- Thalweg of "Estevillas" (or "Los Migueles"), so called because it comes from the late Roman settlement of the same name, which was recently excavated.

The "Los Ahijones" stream flows into the area around "Los Ahijones" and "Los Berrocales", and from the latter, it flows into the "Los Migueles" stream (or "Prado" stream)

(Figure 1). It is a rather flat area, of which approximately 20% is urbanized by the old town of Vicálvaro, the Industrial Estate, and other neighborhoods, such as "El Cañaveral" (539 ha), which began construction work at the end of 2006. This district also has a large amount of land that has not yet been developed: "Los Berrocales", "Los Ahijones", and "Los Cerros".

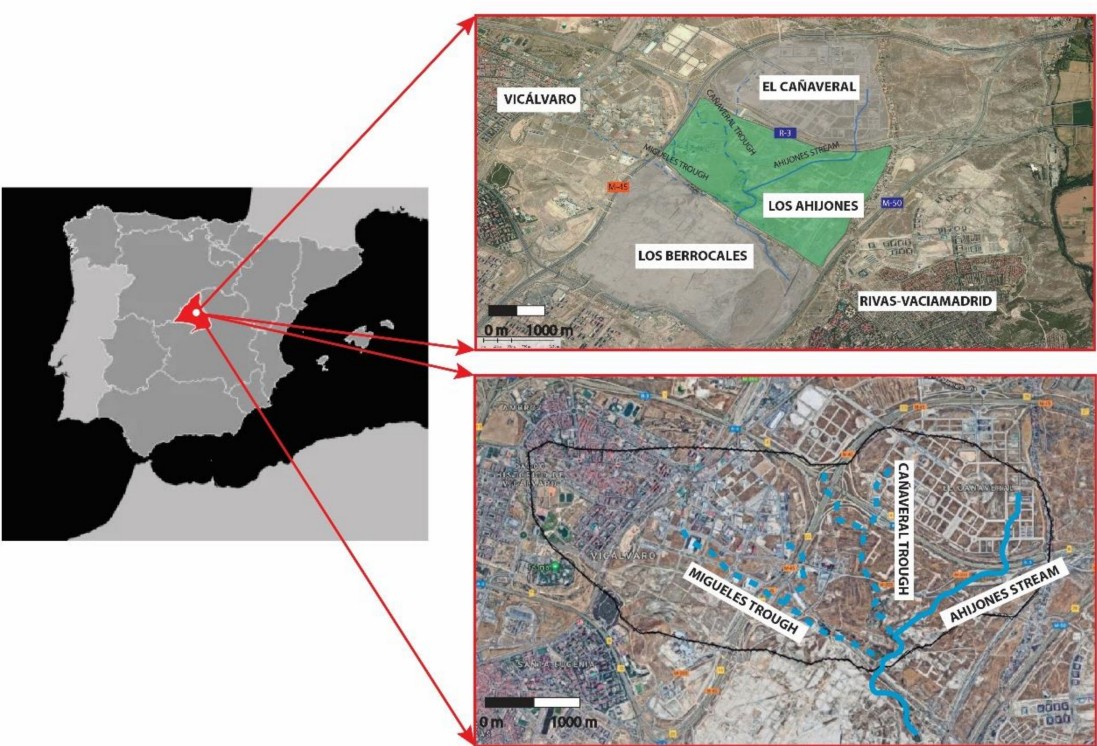

**Figure 1.** The situation of the study area in Vicálvaro (Madrid) is shown in green, the Ahijones urbanization is shown on the left, indicating the hydrological basin of the three watercourses that pass through the urbanization, which are shown on the right.

The study area is located in the southeast area of Madrid, an area of urban expansion and where Miocene clay and gypsum formations are abundant. Specifically, the geology of the site, from the stratigraphic point of view, differentiates a Lower Unit from the Lower-Middle Miocene of about 80 m in visible thickness. Two subunits can be distinguished: more or less massive gypsum at the bottom and gypsum with interspersed layers of clay on top. They do not appear on the surface but have been identified in boreholes. Above is an Intermediate Unit of about 20 to 30 m in total thickness, which is made up, from the bottom up, of clays and green silts, with interspersed layers of fine-to-medium-grained and very micaceous sands, a more permeable layer of sandstone, dolomites, limestones, flint levels, and clay intercalations, and, finally, sepiolitic clays at the base and sepiolites at the roof.

In the Quaternary, three formations can be distinguished: sand paleo-channels adjusted to the main drainage network (no more than 3 m thick), an alluvial from the "Los Migueles" stream (up to 9 m thick), and tilled soils. The excavations carried out have mostly eliminated these surface formations or have broken their lateral continuity, except for the alluvial of the "Los Migueles" stream. While the post-orogenic Miocene deposits are generally arranged horizontally, it is known in Madrid that in the areas directly or indirectly related to the gypsum, the sedimentary cover is folded, so one cannot speak of proper tectonics. The whole depression of the urbanization of "Los Ahijones" is shaped as a depocenter for the Intermediate Unit, and due to the underlying karstification processes

concentrated on a NW-SE fault coinciding with the "Los Migueles" stream, a long and narrow syncline has been produced.

The Miocene formations represent certain inherent geological and hydrogeological risks. One of these risks, associated with the poorly permeable clay-like facies of topographically depressed areas, such as the one in this study, is that the phreatic can be close to the surface, and this can pose a problem during urbanization works and for the subsequent habitability of the urban area. This aspect has not been studied until now, and the following causes have contributed to this:

1.  As they are not exploitable aquifers for supply, they have not been studied, and their hydrogeological functioning is unknown.
2.  There is an administrative difficulty, since it is not compulsory to conduct prior hydrogeological studies.
3.  There is suspicion that there could be problems with the phreatic falls when it is revealed in the surveys of the geotechnical studies of the project. However, as the land is impermeable, the water table may not be visible during the short period of time during which the paving is carried out. This is what happened in our study area, where the appearance of the phreatic was an unexpected circumstance. In fact, and as we have seen during the study, the water table reacts a day later to the opening of the troughs.

In fact, during the earthworks necessary for the excavation of the roads and the opening of trenches for the sewage system, the water table was cut over a wide area, which mainly emerged in the sewage trenches and remained in the form of flooded and waterlogged areas for weeks and months, which significantly hindered the urbanization work.

At this point, the decision was taken to embark on a process of detailed study and gain knowledge of the terrain and its hydrogeology in order to protect the infrastructure being built and the resources that the flow of groundwater involved, abandoning the usual procedures in this type of case based on the temporary evacuation of water during the execution of the works, which are already using known pumping techniques.

There was an urgent need to learn more about the hydrogeology of the site. As it is not an area of aquifers, there were insufficient data, and an extensive campaign of hydrogeological reconnaissance surveys had to be carried out. These surveys, together with the analysis of the hydrogeological information from previous geotechnical studies, provided insight into the conceptual hydrogeological model and made it possible to apply a mathematical flow model. This mathematical model served, in turn, to complete the design of the drainage works, which were successfully executed shortly afterwards.

In this way, it was decided that we would take advantage of the drainage trenches, which were to be dug to install a collector network, in order to attach a lateral drainage network made up of perforated pipes for the collection of groundwater. This network was supported by a thick concrete layer, covered with filtering gravel, and confined by geotextile. The network is completely independent of the sewage network and flows by gravity into the course of the "Los Ahijones" creek at approximately 500 m from the conflict zone, safeguarding the natural resources present in the area.

This study applies a working methodology that we could call an "emergency" methodology, i.e., a methodology for a non-ideal situation, which is seen here as necessary, given the lack of previous hydrogeological studies. However, the fact that it was carried out during the process of carrying out the work has meant that many very complex field trials have been carried out, sometimes not because of the high economic cost involved. This experience has served to raise awareness of the need to carry out prior hydrogeological studies during the initial phases of any urban development.

## 4. Results

### *4.1. Hydrogeological Conceptual Model*

4.1.1. Definition of Hydro-Stratigraphic Units. Permeabilities and Transmissivities.

There are three different hydro-stratigraphic units, which, from top to bottom, are as follows:

- Quaternary surface formations:

The permeability of the paleo-channels and soils ranges from $100^{-10}$ m/day $(1.15–10^{-3–4}$ m/s). For the alluvial, "Los Migueles", and tributary streams, the result is K = 10 m/day $(1.15–10^{-4}$ m/s). The average transmissivity of this last aquifer is T = 50–100 $m^2$/day.

- Miocene formations:

Layer 1. The first layer, or upper aquiclude, is made up of a group of Miocene clays, which, from the bottom to the top, are differentiated: gypsum clays, marly clays, sandstone levels, and sepiolitic clays. This layer has an average thickness of 20–30 m. It is a hydrogeological unit that does not work as a confined aquiclude and is recharged with rainwater.

According to the Lugeon tests, this unit could have a joint average permeability of $10^{-8}$ m/s, even if there are values of $10^{-6–9}$ m/s. The sands and dolomites have a slightly higher permeability of around $10^{-7}$ m/s. For the purposes of simplification, the equivalent horizontal permeability of this layer 1, with a horizontal flow parallel to the strata as it actually occurs, will be the arithmetic weighted average by the thickness of the layers, resulting in a K = $2 \times 10^{-8}$ m/s ($1.67 \times 10^{-3}$ m/day). The values of transmissivity vary between 0 and 0.036 $m^2$/day, that is, very low values, which are typical of aquicludes. This explains the low flow rates in the drainage trench, as water is currently being extracted from the clays.

Layer 2. Underneath, there is a layer of gypsum, also from the Miocene period, which is about 80 m thick. The gypsum has a low matrix permeability (around $10^{-8}$ m/s), but it can be very heterogeneous and depends a lot on its degree of karstification (locally $7.2 \times 10^{-5}$ m/s in the gypsum-clay contact).

4.1.2. Geometry and Limits of the Clay-Like Miocene "Los Migueles" Stream Fault Zone

Taking advantage of the data from the numerous soundings and pits available, the isoline maps of absolute dimensions and contact depths of the layers defined above were generated. With these maps, it was possible to make a block diagram (Figure 2) and geological section representative of the urbanization work (Figure 3). We can see in these figures how the layers are sub-horizontal, although they adapt to a dissolution depression, which affects the casts, giving rise to a syncline and a fault. The "Los Ahijones" depression seems to be shaped as a depocenter, where the underlying karstification processes, concentrated in a NW-SE fault and coinciding with the "Los Migueles" stream, have created a long and narrow syncline in the clay-like cover, which will condition—in an important way—the underground drainage, as we will see (Figures 2 and 3).

This fault fits into the course of the "Los Migueles" stream, and, although it was not known, it must be included in the group of faults that affect the substrate in this part of the Tagus basin and which also end up being reflected in the cover. This local re-sinking has preserved the entire stratigraphic column from erosion, including the Quaternary. This fault is important; it is of the direct or normal type, measuring a jump of about 5–10 m, and the sunken block is the northern one. We can see a smoother syncline to the north, halfway up the slope, which is very well reflected in the isolines at the base of the guide layer of sandstones (Figure 2).

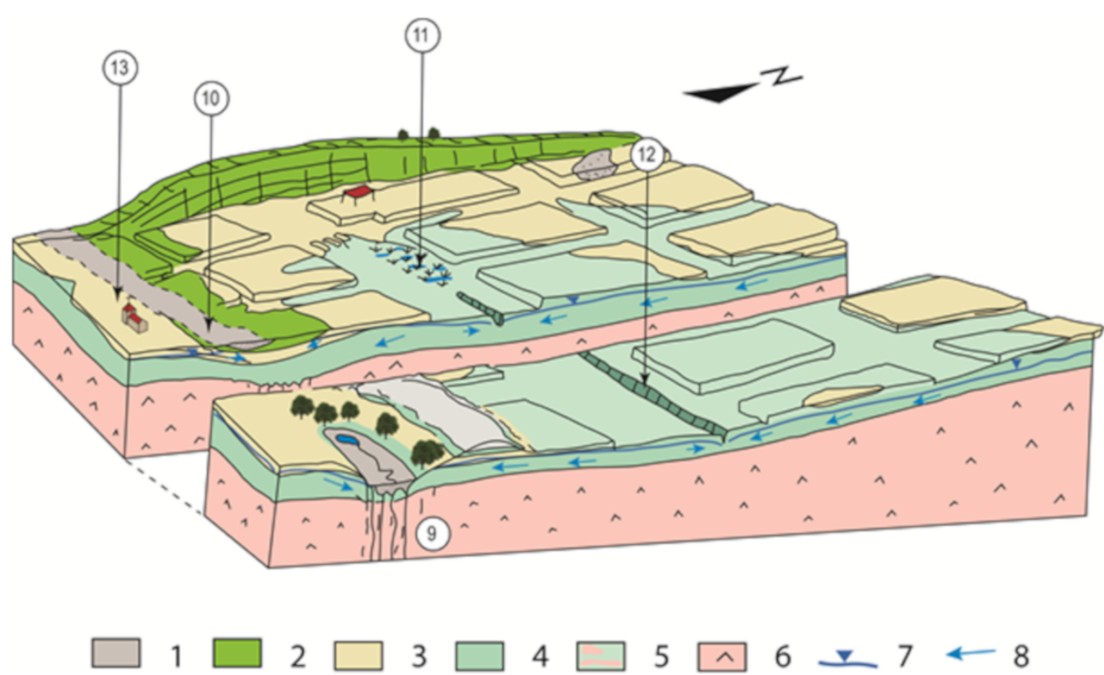

**Figure 2.** Geological block diagram of the urbanization of "Los Ahijones" during the road excavation works (July 2019) (without scale). Quaternary: 1. Alluvial of the stream. Miocene: 2. Sepiolitic clays. 3. Sands, chert nodules and dolomites. 4. Green clays and silts ("peñuela"). 5. Clays and green silts with gypsum intersperses. 6. Massive gypsum. 7. Water table. 8. Underground flow. 9. Fault zone. 10. "Los Migueles" thalweg. 11. Waterlogged areas. 12. Drainage trench. 13. "Nuestra Señora de la Torre" Hermitage.

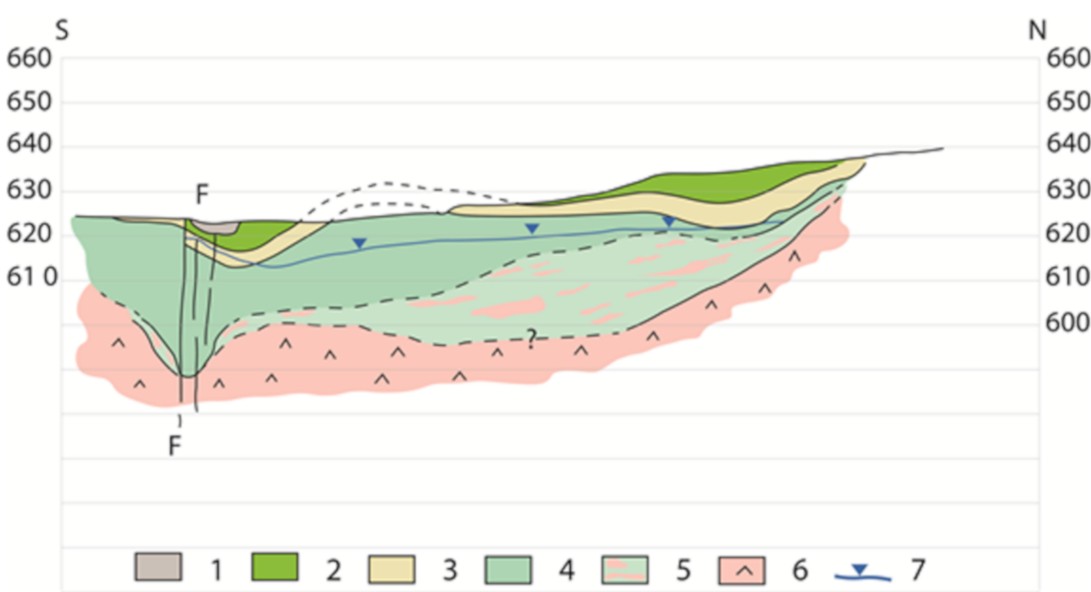

**Figure 3.** North–south representative geological section of the urbanization of "Los Ahijones". Quaternary: 1. Alluvial of the stream. Miocene: 2. Sepiolitic clays. 3. Sands, chert nodules, and dolomites. 4. Green clays and silts ("peñuela"). 5. Clays and green silts with gypsum intersperses. 6. Massive gypsum. 7. Water table. F. Fault zone.

In the fault zone, higher permeabilities must be considered for all the Miocene lands affected by it but which are not known beforehand. This is one of the cases in which modelling has been helpful. From experience in the area, it is known that green clays, if fractured in a smooth form, can greatly increase their permeability. This has been proven during the excavation of the Los Migueles storm tank, an infrastructure work

built to laminate the rainwater from all the sectors of the south-east ("El Cañaveral", "Los Berrocales", "Los Ahijones" and "Los Cerros"), where another fault was cut parallel to ours and mapped on the 1/50,000 Madrid geological sheet [17], and where most of the groundwater flowed (oral communication by Felix Escolano, senior geologist of the work).

- Syncline permeability

The elongated syncline NO–SE, adjusted to the fault that has given rise to it, was assigned a permeability value of $5 \times 10^{-5}$ m/s, which results from the estimation of an equivalent hydraulic conductivity, in which not only the clays of "peñuelas", but also sepiolitic clays, the more permeable sandstones of $10^{-7}$ m/s, and a 3 m top layer of alluvial of $10^{-4}$ m/s intervene. However, it must also be taken into account that these materials (except the alluvial ones) are crossed by a major fault line. Syncline does not affect layer 2, but the removal of this layer by dissolution has deformed layer 1 by sinking it into the syncline. However, layer 1 is crossed by the fault, and this will increase its karstification and therefore its permeability.

For modelling purposes, layers 1 and 2 will be considered, as well as the alluvial of the Los Migueles stream (the rest of the Quaternary is considered not to come into play). The geometry in the syncline and the mentioned fault will be considered. The hydrogeological limits of the clay-like Miocene should in principle be limited to the hydrological basin, since, as we can see, the phreatic surface is very adapted to the topography.

### 4.2. Natural Recharge

Before the beginning of the urban development processes and the subsequent excavations, the aquiclude was mainly recharged by the infiltration of rainwater through the soil and permeable quaternary surface formations, such as the paleochannels. Since this clay-like Miocene (layer 1) retained the downward flow at the base of the working soil, the evaporation and evapotranspiration of these cereal fields must have been an important outlet in the hydraulic balance in the past.

For the modelling, an estimation of the natural recharge values in the Miocene aquiclude of the area was made by means of a water balance of an average year using the average monthly rainfall (P) of station n° 39 in El Retiro (Madrid) [18], which is relatively close to the study area, and also using the potential evapotranspiration (PTE) of Madrid. It is thus estimated that there is an excess of water (recharge + surface runoff) of 16 mm, which accounts for 3.7% of precipitation, with most of the rainwater being consumed in evapotranspiration. Recharging takes place in the winter and spring months of February and March, when evaporation is lower. Normally, given the impermeability of layer 1 below the soil, most of the excess water is used for surface runoff, especially now that most of the soil has been removed.

If we consider that the inputs (natural recharge) to the aquiclude are equal to the outputs (discharges), with more or less variations in storage, we can say that for the purposes of visualizing the little value of the natural recharge of this area, in eight months of 2019 (April-December), no natural springs have been observed. The discharge of the system was probably verified by the EVT of the vegetation and also, in a certain portion, of the drainage of the ditch (1 L/s), since this has been supposed as an emptying point of the aquiclude. Extractions of groundwater by means of pumping wells were (and are) negligible. All these uncertainties make it necessary to address the issue with a mathematical hydrogeological model.

### 4.3. Aquiclude Discharge: The Drainage Ditch and Lengths of Influence

Apart from the flow of the aquiclude directed to the alluvial of the "Los Migueles" stream, the discharge of the clays is carried out through a drainage trench about 3 m deep and one kilometer long. The purpose of this trench was to lower the water table in those areas where it was above the sanitation network and which included, above all, the central area.

It was designed with a layout that was adjusted on one side to this area of the high phreatic, taking advantage of the central road of the urbanization with an east–west direction and perpendicular to the flow line directions, a good criterion for the efficient capture of the flow from the north, although it also captured a strip of the southern area in its radius of influence. Other perpendicular roads start from this trench towards the north, which in turn fork into other smaller ones in the shape of a fishbone. It is intended to continue its extension and development by other roads, until it covers the area of the high water table.

The trench is mostly dug in green clays ("peñuela"), as the upper layers were removed during the excavation or are now hung up high between the roads in areas intended for building plots. The trench constitutes an aquiclude with a uniform recharge and presents a natural flow from north to south, with a variable hydraulic gradient of about 0.25%. Figure 4 shows the initial phreatic level (approx. 617 m) and the depression curves of the dynamic phreatic level on both sides of the trench and on two dates separated by about 10 days. Firstly, as expected, we can see that the length of influence of the side from which the natural flow comes (north side) is greater than that of the south. To the north, the length of influence can be very large, certainly more than 100 m, and increases with time, logically. To the south, it is smaller and does not reach 25–30 m. It can be seen that the trench has lowered the water table very effectively in a short time, although as the terrain is very impermeable, it is difficult to empty it.

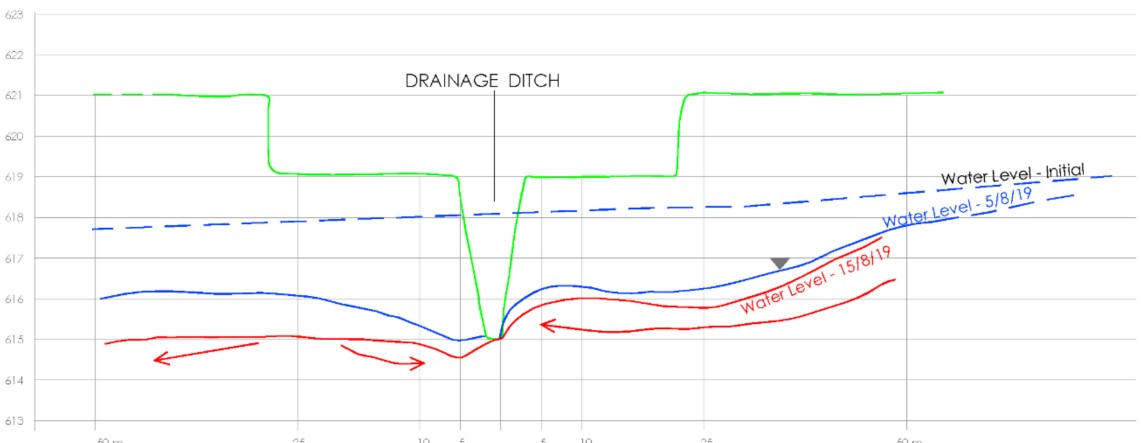

**Figure 4.** North–south transversal profile to the drainage trench, with the initial water table situation, descent curves, and distances of influence.

The flow that has drained the trench has been measured during the months of July and August 2019 in its final drainage. The average flow is small (around 1 L/s), as water is being drawn from the clays and comes mostly from the central lower trench. If we obviate the possible effect of the initial emptying of the first measurements, we can establish the following exhaustion curve in an uninfluenced regime and take advantage of the month of August without rain, where $Q_0$ = 1.12; Q = 0.7 L/s; and t = 21 days. The depletion curve is as follows:

$$\alpha = (LnQ_0 - LnQ)/t = (4.5 - 4.1)/21 \cong 0.02 \text{ days}^{-1} \tag{1}$$

where $Q_0$ is the flow rate at the beginning of the depletion curve, Q is any flow rate within the depletion curve, t is the time between both flow rate measurements, and $\alpha$ is the depletion coefficient.

Knowing the value of $\alpha$, it follows that for the drainage flow of the trench to be reduced to a negligible value (0.1 L/s, for example), it would take a time of:

$$t = (LnQ_0 - LnQ)/\alpha = (4.5 - 2.1)/0.02 \cong 120 \text{ days (4 months)} \tag{2}$$

In the main section of the trench, water is calcium-magnesium bicarbonate, with a conductivity of 1150 µS/cm, which is typical of the "peñuelas" without interspersed gypsums, while water in the final section of the "Los Ahijones" creek is already sulphated, with greater conductivity (3000 and 6000 µS/cm), revealing the proximity of the gypsums to the surface, although they do not emerge. The temperature of the underground water varies between 18 and 16.5°. The pH is always 7.9. In the end, the water comes out of the trench with 1334 µS/cm of conductivity and calcium-magnesium bicarbonate facies, that is, water suitable for irrigation.

### 4.4. The Underground Flow

#### 4.4.1. Approach to the Underground Flow System before the Works in "Los Ahijones"

The alluvial and fractured clays of layer 1 around the "Los Migueles" stream (or "Los Prados" stream) formed an aquifer blanket of local interest, which fed the riverbed as a gaining channel through springs with a few L/s flows. It is now a zone of hidden discharge from the Miocene through the Holocene alluvial sediments and coincides with the fault zone and bottom of the syncline. There were also about 15 shallow and manually dug wells, given the proximity to the surface of the phreatic, at the bottom of this thalweg. They were used for watering gardens and domestic needs, but most of them have been abandoned for 40 years. The flow of the wells was around 60 m$^3$/day. The water was potable, as the flow had not yet circulated through the gypsum layers, which would naturally contaminate it with sulphates.

At the point where this stream crosses the "Los Ahijones" stream, the water table emerges in permanent pools, and after very rainy periods, water springs up, which we also noticed in 2020.

Figure 5 shows a water table map (contours in masl), from the central area of the urbanization, without underground water extraction wells and without the urbanized Cañaveral district. This is an approximation of the natural regime. The concentration of the flow around the "Los Migueles" stream is striking.

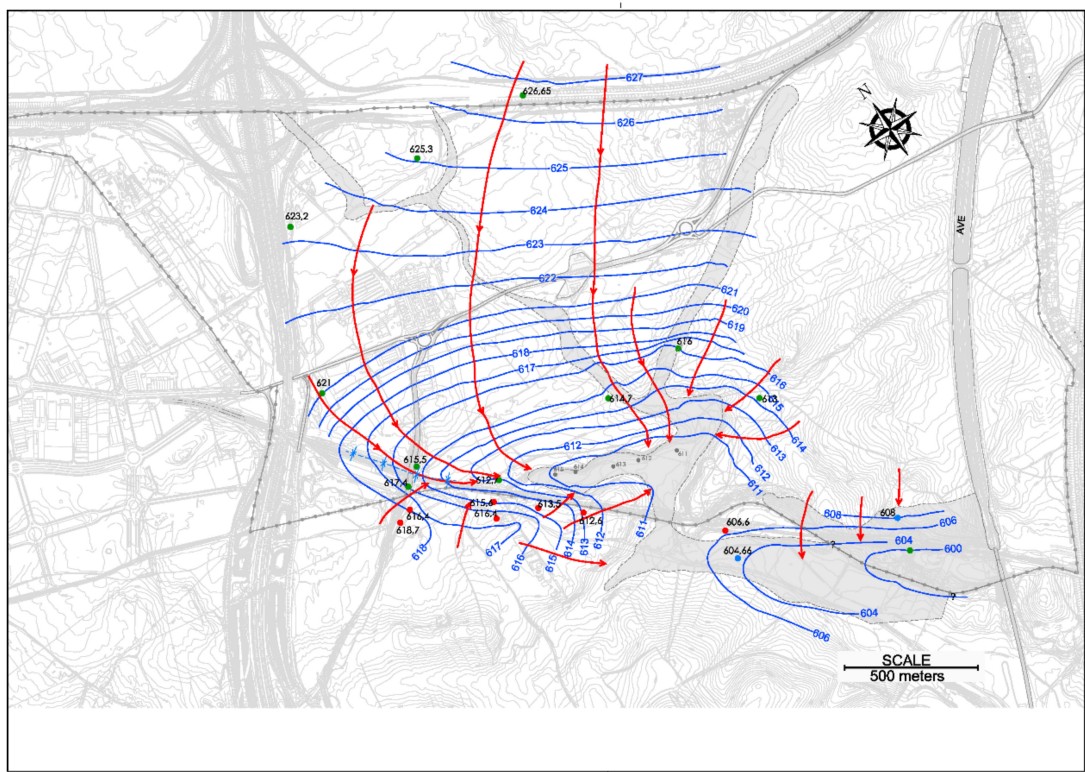

**Figure 5.** 2006 water table map (contours in masl), before the "Los Ahijones" urbanization works.

4.4.2. Hydrogeological Conceptual Model

Figure 6 shows a water table map (contours in masl) from the central zone of the urbanization work, corresponding to July–August 2019, with the drainage trench of one kilometer in length and part of the urbanization excavations. "El Cañaveral" neighborhood has already been urbanized. It follows the trend of the concentration of the flow in the "Los Migueles" stream, although the influence of the drainage trench is noticeable.

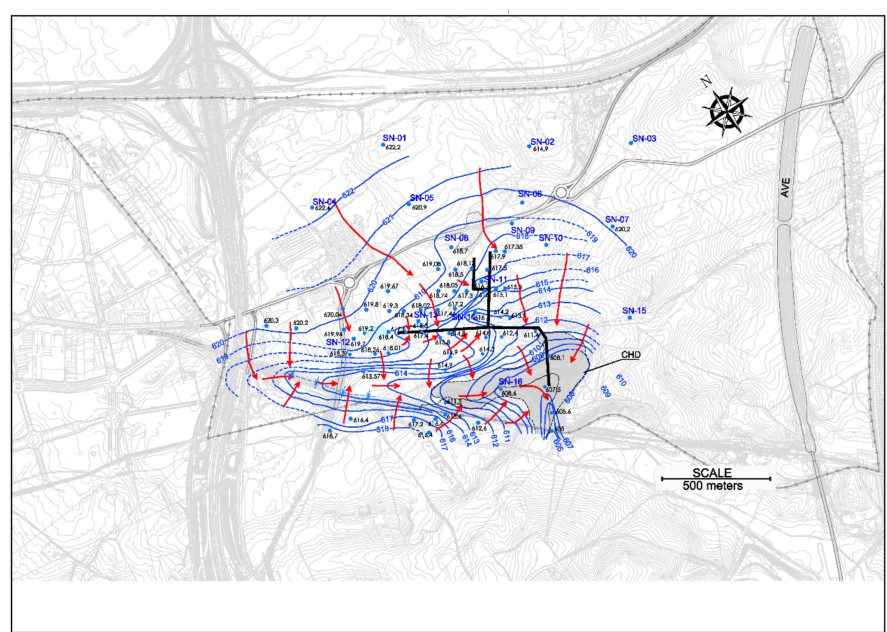

**Figure 6.** 2019 Water table map (contours in masl) during the "Los Ahijones" urbanization works and the beginning of the excavation of the 1 km-long drainage trench.

*4.5. Hydrogeochemistry*

4.5.1. Types of Hydrochemical Facies and Their Spatial and Depth Zoning

There are two quite clear chemical facies: a calcium-magnesium bicarbonated water facies, which extends in the area west of the "Los Ahijones" stream. Another one of calcium sulphate facies is located approximately on the axis and east of the aforementioned stream (Figure 7a). Figure 7b shows Piper's diagram, presenting the two types of hydrogeochemical facies.

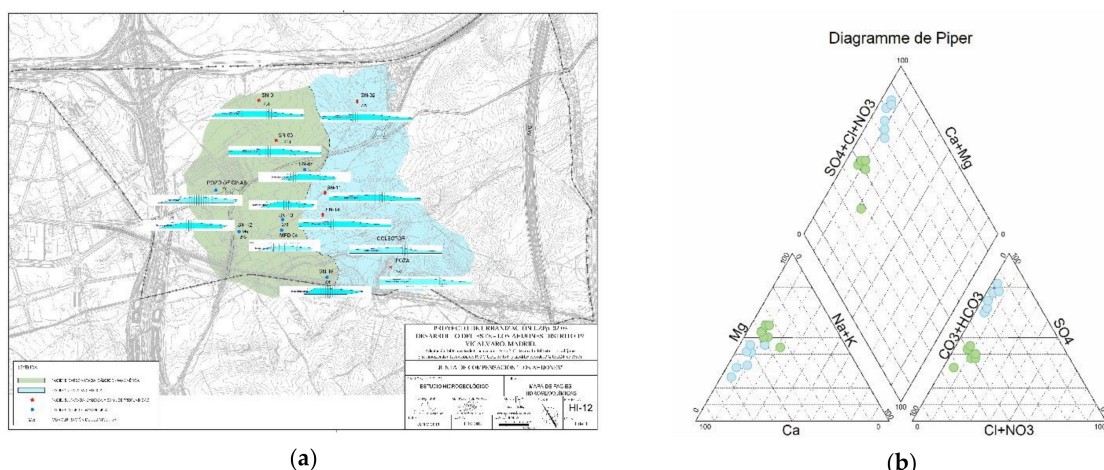

(**a**)  (**b**)

**Figure 7.** (**a**) Distribution of the hydrochemical facies of the groundwater and Stiff diagrams: Sodium-magnesium bicarbonate facies (in green); and calcium sulphate facies (in blue). (**b**) Piper diagram showing the two families of waters: sulphate waters (in blue); and sodium-magnesium bicarbonate waters (in green).

The calcium-magnesium bicarbonate facies are associated with the intermediate Miocene clay units. The waters are hard, and their high magnesium content is probably due to the presence of magnesian clays. It is the type of water that predominates on the surface when there is no presence of gypsum near it. Where the casts are more than 5 m deep, the first of the facies usually predominates, and where the casts are shallower, the second predominates. However, it can also be observed that the content of plaster must increase towards the east due to the lateral change of facies within the clays. On the other hand, the analyses have not detected an excessive content of chlorides that would lead us to suspect the existence of hyper-soluble materials. The distribution of the conductivities reflects how they increase in those areas with more sulphated water and how the sulphate content adjusts to the geology of the area near the gypsum layers.

On the other hand, the pH value of the water has a value that oscillates in all cases in a narrow range around pH = 7.9. The temperature of the waters is mostly between 18° and 20°, and in the hydrodynamically most active area of the syncline of the chapel, it drops to 15–16°.

### 4.5.2. Isotopic Hydrochemistry

Seven representative samples were taken, which are spatially distributed throughout the development, corresponding to shallow groundwater (drainage trench, alluvial pool, etc.) and others somewhat deeper (piezometers). The results of the isotopic content of $^{18}$O and D are presented in Table 1. We observe that the narrow range of values ($^{18}$O between 6.50 and 7.15, D between 47.8 and 51.0) indicates the same origin of natural recharge in the whole area. The values for the content of $^{18}$O and D of all the samples are very similar to those typical of rainwater in the winter period, in which there is natural recharge in Madrid (-averages of 7.73 $^{18}$O, $-53.7$ D) [19]. In other words, the recharge is from rainwater. Both the shallower groundwater (values of $^{18}$O less than 7.0, and values of D less than 50.0) and the slightly deeper piezometers ($^{18}$O slightly more than 7.0, and D slightly more than 50.0) are similar to each other, indicating the same period of natural recharge for each. This small difference between both is logical, since the origin of the recharge rains corresponds to different periods, which are older in the deeper ones. In other words, there is a certain isotopic stratification of the waters in the vertical.

**Table 1.** $^{18}$O and Deuterium content of the representative groundwater samples from "Los Ahijones".

| Sample Reference | $\delta^{18}$ OVSMOW | $\delta$ DVSMOW |
|---|---|---|
| P.SN 11 | −7.04 | −50.0 |
| BACKWATER SUPPLY | −6.50 | −47.8 |
| TORRE PEDROSA WELL | −6.62 | −48.6 |
| G 13 | −7.11 | −50.8 |
| P. SN 5 | −7.15 | −51.0 |
| COLECCTOR AUXILIAR | −6.79 | −49.3 |

## 5. Numerical Simulation

### 5.1. Coded Used

The site can be represented by a three-dimensional finite-difference grid, in which the system of differential equations of groundwater flow can be solved. Each of these meshes has its own geometrical and hydrogeological characteristics that intervene in the equation.

The mathematical code used is Modflow 2005 [16], a numerical code that uses the three-dimensional finite-difference method to simulate the flow in the saturated zone. It was developed by the United States Geological Survey and makes it possible to work with anisotropic, heterogeneous, multilayer, confined or free, and porous aquifers. The flow regime can be stationary or transitory.

The ModelMuse program for the Modflow-2005 version is used as a graphical analysis interface.

### 5.2. Geometry of the Model and Discretization of the Mesh

The domain of the hydrogeological model of the Ahijones has been defined according to the hydrological basin of the "Los Ahijones" stream (Figure 1). The area modeled in the model occupies an area of approximately 1309 km$^2$ (13,088,130 m$^2$). It is discretized in two horizontal layers. The mesh size used in both layers is 50 × 50 m, with the difference that in the 2019 model a refinement of 2 × 2 m is carried out along the drainage system in order to achieve greater precision in the simulation.

The model considers the two differentiated layers mentioned above: Layer 1. Upper aquiclude of Miocene clays, with an average thickness of 20–30 m; and Layer 2. Lower aquiclude of plaster, with a thickness of about 80 m. The geometry of the contact is perfectly defined in 3-D. The base of the model has been placed at a depth of 70 m with respect to its stratigraphic ceiling, which corresponds approximately to its thickness.

As previously explained, one of the main differences between the modelling of 2006 and 2019 is the variation in the topography, as in the latter year, there have been significant changes due to the urbanization works.

### 5.3. Hydrogeological Parameters

Once the geometry of the model has been defined, the hydrogeological parameters of the model are established. In Table 2, the parameters, which have been introduced in the model for both layers, are shown. The values of horizontal permeability have been taken mainly from the Lefranc tests, although other permeability tests have also been taken into account.

**Table 2.** Hydrogeological parameters.

|  | Permeability kx | Permeability ky | Permeability kz | Horizontal Anisotropy | Water Table Initial |
|---|---|---|---|---|---|
| Layer 1 shallow aquiclude | 1E$^{-8}$ m/s | 1E$^{-8}$ m/s | Kx/10 | 1 | Topography-1 |
| Layer 2 Deep aquiclude | 1E$^{-8}$ m/s | 1E$^{-8}$ m/s | Kx/10 | 1 | Topography-1 |

### 5.4. Outline Conditions

1.  The "Changing Head Defined" (CHD) condition is used to define a fixed and known piezometric height. This condition is established at the limit between the alluvial of the "Ahijones" stream at its exit from the urbanization work and its contact with the clay-like Miocene on the edge and gypsum on the bottom. It is a permeable border with free water or a highly permeable medium, as is the case, where the contrast of permeabilities from $10^{-4}$ to $10^{-8}$ is very great. Here, the potential is constant, and the 612 m equipotential has been set as the representative of this side, which receives the current lines. Indeed, this condition is undoubtedly supported by the fact that the equipotentials outline the exterior of this alluvial and the lines flow along this edge and are perpendicular to it. To better follow this reasoning, the indication of the CHD is included in Figure 6.
2.  Concerning the natural recharge used in the model, it has already been commented on above.
3.  The piezometric level observation points are as follows: Eight piezometers have been used to calibrate the 2006 model, and seven have been used for the 2019 model. Their location is shown in Figures 5 and 6.

### 5.5. Model Calibration

The simulation of the 2006 model resulted in the water table map presented in Figure 8 for layer 1, where the direction of the groundwater flow is highly conditioned by the fault and the axis of the elongated syncline, clearly confirming the conceptual hydrogeological model of the aquiclude. The distribution of equipotentials in layer 2 is similar. The model is very sensitive to the permeability of this narrow corridor, and a permeability of

$5 \times 10^{-5}$ m/s had to be assigned to obtain a good calibration. The same syncline in the layer corresponding to the massive gypsum has been assigned a permeability of $1 \times 10^{-7}$ m/s.

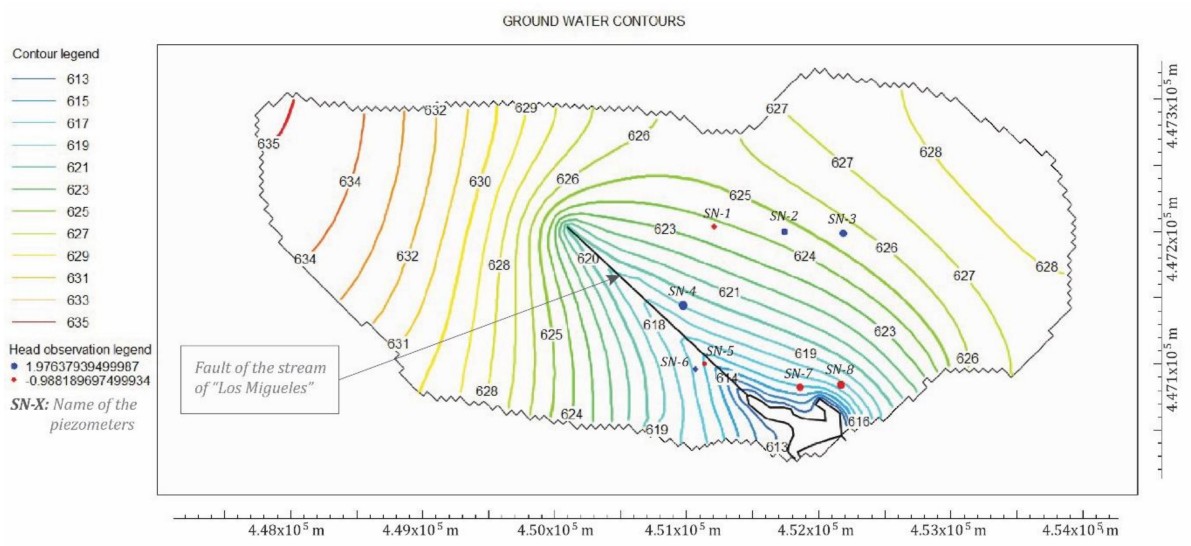

**Figure 8.** Steady state simulation: equipotential lines for the 2006 conditions, including the preferential flow path associated with the syncline.

Figure 9 shows the adjustment or calibration line for each of the observation points, and as can be seen, all the points have low residual errors, as they are very well adjusted to this line. As a representative statistical indicator of the degree of adjustment, the mean square error corresponding to all the observation points is used. This is 1.14 in the 2006 model, which is considered a fairly low error.

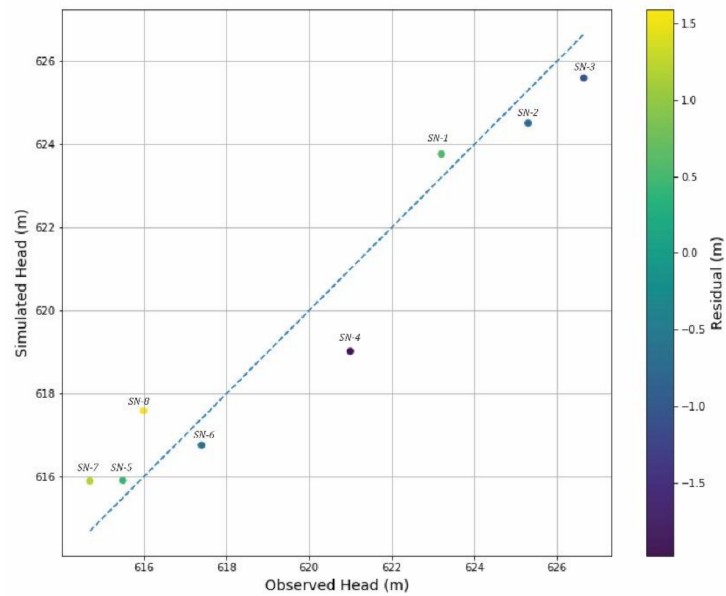

**Figure 9.** Calibration line of the different observation points for 2006.

The simulation and calibration of the mathematical model in 2019 showed, in addition to the variation in the topography due to the work carried out, the incorporation of the existing drainage and the reduction in the recharge in the area of "El Cañaveral", which represents 20–40% of the original recharge.

The results of the model simulation during the 2019 low water season are shown in Figure 10.

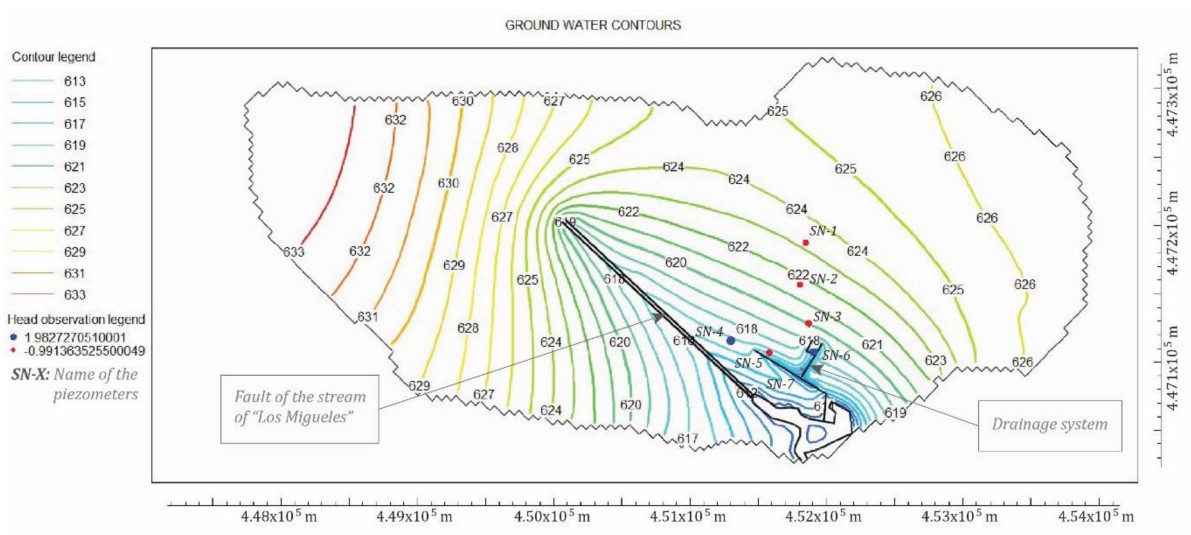

**Figure 10.** Steady state simulation: equipotential lines corresponding to the 2019 condition.

It can be seen that the general flow pattern of 2006 is maintained, although there has been a general decrease in the equipotential lines in the model, especially in the area of "El Cañaveral", with decreases of nearly two meters.

The line of fit (Figure 11) and the RMS error of all observation points show that the 2019 model has also been successfully calibrated, presenting an RMS error of 1.199.

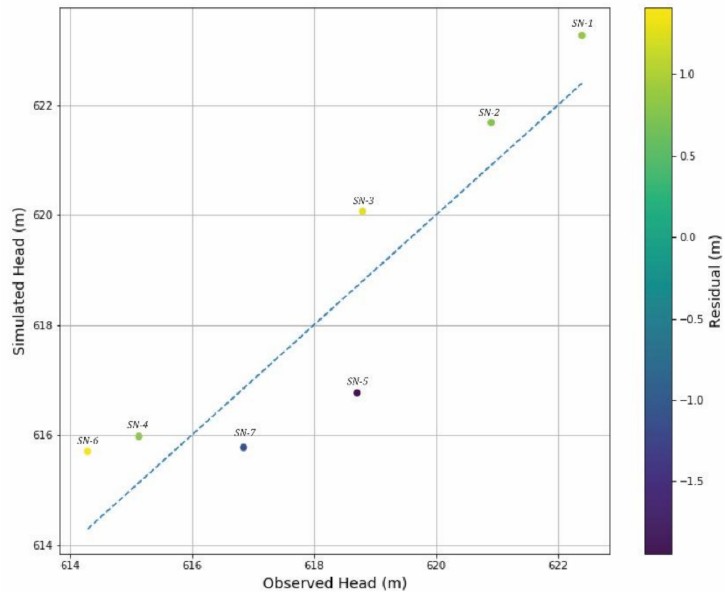

**Figure 11.** Calibration line of the different observation points for 2006.

*5.6. Results of Different Simulation Scenarios of the Calibrated Model*

Once the mathematical model has been calibrated, different simulations are carried out in order to analyze different scenarios.

- Simulation in a transitional regime during 2019, with drainage

We analyzed the variation of the phreatic over time using a transitional model of 100 days, with 4 periods (each period having 25 days). As can be seen in Figure 12, the

drains create a depression in layer 1 of up to 2 and 3 m of the phreatic around it, notably modifying the nearest equipotential lines.

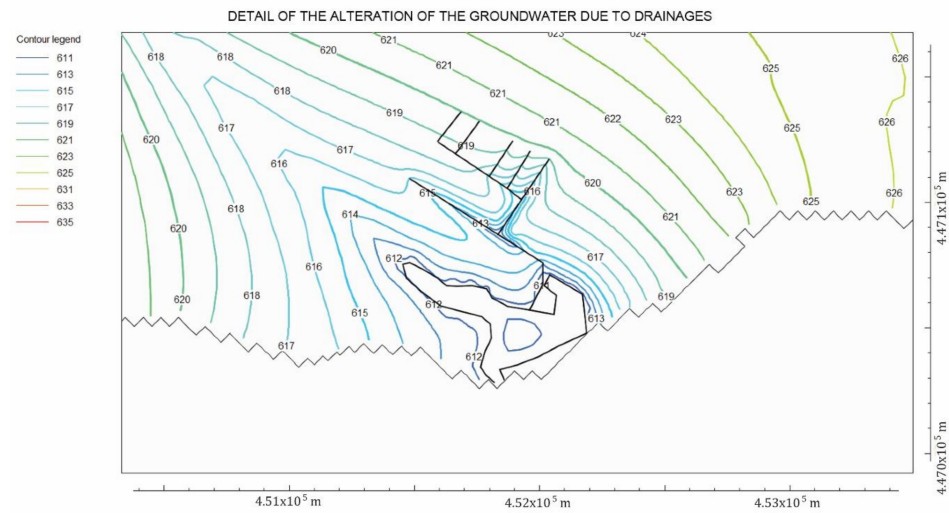

**Figure 12.** Simulation in a transitional regime: alteration of the equipotential lines of layer 1 due to the drainage imposed in 2019.

It can be seen that, in this transitional model of 100 days, the general equipotential lines of the model remain intact but are not closest to the drains, which have suffered significant drops in the water table. The average velocity of the groundwater, considering an effective porosity of $m_e = 10^{-3}$, $K = 10^{-8}$ m/s, and an average gradient of 0.25%, is 7.84 m/year, which is very low and reveals a very impermeable terrain, typical of clay. Near the trenches, the speed can increase significantly as the gradient increases. On the other hand, the effect of the drains on the flow through the massive plasters is null.

On the other hand, the section of the model shown in Figure 13 reflects the good role that the drains have played in this transitional model and that they reflect the reality, which is that the drains have achieved a drop of two to three meters in the water table in their vicinity, managing to depress the water table in the flooded areas. The red line corresponds to the situation of the water table prior to the installation of the drains, and the blue line corresponds to the water table 100 days after the drains have been installed.

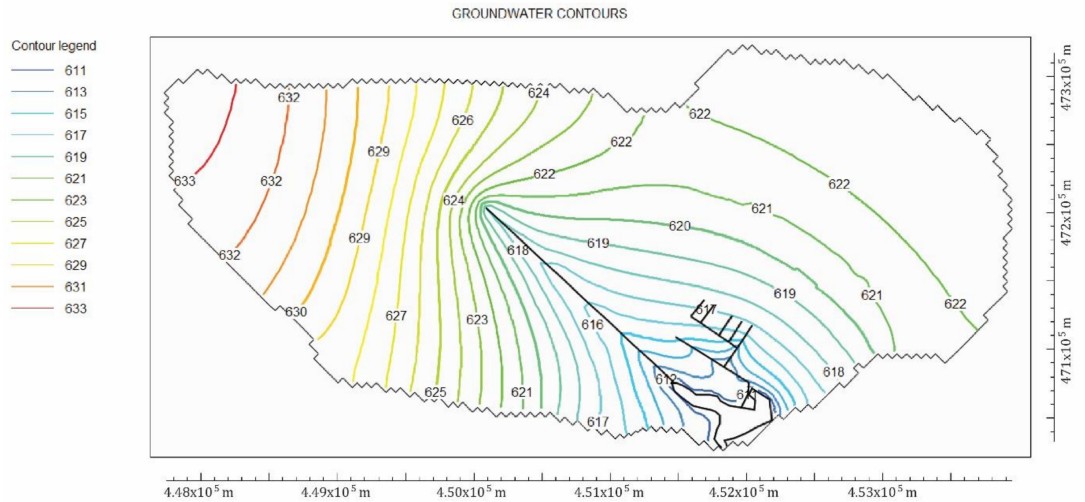

**Figure 13.** Vertical section of the model showing the equipotential lines and the water table obtained after 100 days of drainage. The water table obtained in a simulation without drainage is also shown (the red line indicates the roof of the plaster).

- Comparison of the stationary simulation results, with and without drainage (future model).

The effect of the drains is simulated in a stationary state, i.e., in the long term. The objective is to see whether the drains will have a direct effect on the whole model, or whether their effect will only be local. The results indicate that the greatest decreases occur logically in "Los Ahijones" and that the effect will be up to 1 m in the area corresponding to "El Cañaveral". The decrease is relatively insignificant in the area corresponding to Vicálvaro.

- Stationary simulation for a future scenario with less recharging.

A future hypothetical situation is that of the complete urbanization of "Los Ahijones", which we have assumed will reduce the recharge to 40%. In this situation, the isopiestic lines decrease globally by about 4–5 m, although the general flow pattern remains the same as in the predecessor models (Figure 14).

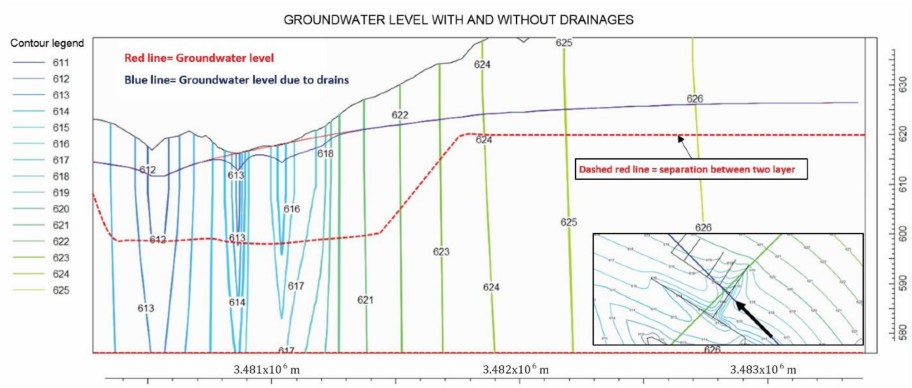

**Figure 14.** Stationary equipotential lines obtained for a 40% decrease in recharge.

## 6. Discussion

In this area, the soils are very permeable, and as we have seen, the phreatic conforms quite well to the topography, with a rather epidermal flow, which is associated with the clay-like formations of the Miocene. The syncline and fault of the winning "Los Migueles" stream redirects the underground flow in a west–east direction, a true depression of the phreatic. The topographic spur where the "Nuestra Señora de la Torre" hermitage is located acts as an underground dividing line with the area of "Los Berrocales" (Figures 2 and 3).

The conceptual hydrogeochemical model can be explained very well by the geology of the site: in the Miocene clay layers, there is a predominance of calcium-magnesium bicarbonate facies and also calcium sulphate, as there is an influence of the casts when they are close in terms of depth or located in the areas to the East. From this, we can also deduce that the underground flow is very shallow and does not penetrate much into the depths, because the influence of the casts would otherwise be more noticeable, especially in the recharging areas. In the discharge area of the "Los Migueles" alluvial deposit, it seems that deeper and longer flows appear. A practical consequence of this is that the drainage ditches should not penetrate too deeply into the aquiclude in order not to capture the poorer quality sulphated water.

The mathematical modelling of the underground flow in the Miocene clay-like aquiclude of the "Los Ahijones" hydrogeological basin through MODFLOW 2005 clearly confirms the validity of the hydrogeological and hydrogeochemical conceptual model, both in the natural regime prior to the works and in the regime influenced by excavations and drainage ditches. The differences in elevation between the observed equipotential lines and those resulting from the model are very small.

The development of the model has made it possible to confirm the permeabilities initially estimated for the different hydro-stratigraphic layers and the role played by the

elongated and failed syncline of NW–SE direction. The greater permeability that conditions and redirects the regional flow to a great extent is also highlighted. The drop in the water table at "Los Ahijones" due to the excavations and ditches, compared to the regime before the works, has been about 2 or 3 m in the area covered by the drainage system. The natural recharge in the clay-like aquiclude is very small and does not reach 1% of the useful water in the hydraulic balance (0.16 mm/year). A scenario analysis has been carried out to estimate the future evolution of piezometric levels in the face of changes in land use.

The modelling has provided general knowledge of the hydrogeological behavior of the lower plaster layer. This layer has a general permeability that is similar to the clay-like aquiclude (despite the possible local karstification), and it is recharged in an induced way through this layer. While this supposes a very small flow, on geological time scales, it has certainly meant a slow dissolution of the gypsum layers interspersed between the clays, which has led to the formation of a general macro-depression in "Los Ahijones" and in the axis of the syncline-failure in particular.

It is discharged into the lower alluvial part of the "Los Ahijones" stream, which explains the high sulphate content in the groundwater of this area.

The modelling identifies the areas where the water table appears in the works prior to the excavation of the trenches, thus justifying the suitability of and the need to carry out these drainages in order to be able to execute the urbanization works.

Looking ahead, where much of the recharge at "El Cañaveral" and "Los Ahijones" will be reduced, the results of the model indicate that equipotential lines in these sectors will be depressed, but the general flow pattern will be preserved, with groundwater being channeled and distributed through the elongated syncline. If even the Vicálvaro Industrial Estate were to be urbanized, the general trend would be maintained, even if the phreatic level dropped, and most of the saturated area would remain.

The drainage trench has effectively lowered the water table in a relatively short time during 2019 in those areas where the water table was above the sanitation network. However, this lowering has only resulted in a drained flow of no more than 2 L/s in low water. During the year following 2020, when the rest of the drainage ditches will be completed, and using the piezometric control network, the hypotheses of a slow lowering of the phreatic level around the ditch will be confirmed, although in remote areas, it will rise slightly due to the effect of rainfall (Figure 15). Additionally, the monitoring of the flow of the ditch continues to ensure its effectiveness in rainy periods (Figure 4).

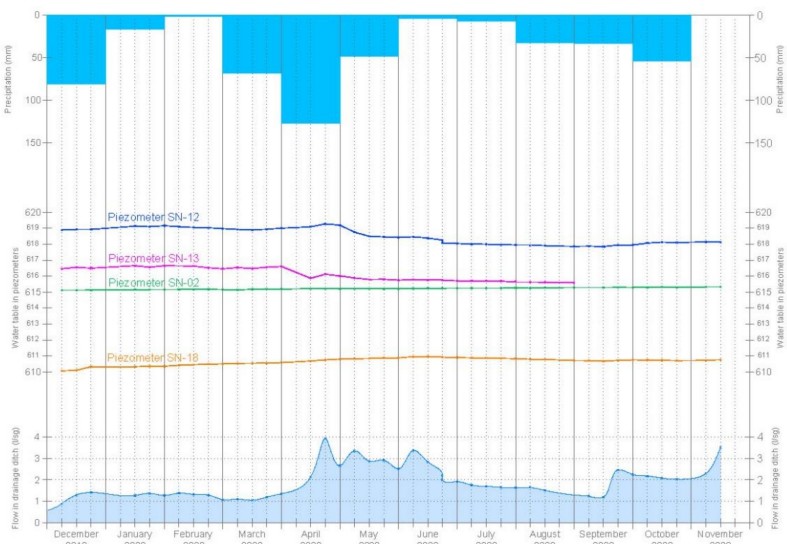

**Figure 15.** Evolution of the volume of water discharged from the drainage ditch observed during 2020. Variation in the levels in the piezometers near to (SN-12 and SN-13) and far from the trench (SN-2 and SN-18).

## 7. Conclusions

The history of construction in large cities is full of examples of incidents of a hydrogeological nature. These problems must be integrated as part of sustainable urban development and for a few years now, there has been an increasing amount of work on this subject, creating a real body of knowledge within the science of hydrogeology, for example, [20,21].

In this work, we offer an case from the city of Madrid, where it is shown how even in arid and impermeable clay areas, we can find complex situations of underground water drainage for urbanization works. It is not always easy to understand the complex hydrogeological functioning of these formations, without previous background and studies. In this case, there is also the peculiarity of a subsoil presenting with other geological complications, such as folds and faults that condition the underground flow.

From the point of view of sustainability, we would like to highlight the ecological benefits of this trench, as it provides the "Los Ahijones" stream with a permanent flow of good quality water. It is a way of safeguarding the natural underground water resources present in the area. The good quality of the water drained from it could be used for irrigation and environmental actions, such as the creation of a small artificial lake in this drainage area for parks.

**Author Contributions:** J.S.d.O., Apply MODFLOW; E.S., field research, conceptual model, and writing—original draft preparation; F.J.E., manuscript supervision, review, and editing; C.S.R., project administration and field research; M.d.P.L., supervision. All authors have read and agreed to the published version of the manuscript.

**Funding:** This work was funded by two research agreements with the Compensation Board Los Ahijones and the Agustin de Betancourt Foundation (Estudio hidrogeológico y del drenaje de aguas subterráneas de la urbanización UZPp02.03 Los Ahijones. Distrito 19. Vicálvaro (Madrid) y modelización matemática del flujo subterráneo en el acuífero/acuitardo de la urbanización UZPp.02.03 Los Ahijones. Distrito 19. Vicálvaro (Madrid) y su entorno.

**Acknowledgments:** The authors express their sincere gratitude to the experts who participated in this research: Felix Escolano (Universidad Politécnica de Madrid). The authors likewise thank Alvaro Sanz de Ojeda from the Madrid School of Mines for his computer support and field work. We also appreciate the work of the three anonymous reviewers who have contributed to improving this paper.

**Conflicts of Interest:** The funders had no role in the design of the study; in the collection, analyses, or interpretation of data; in the writing of the manuscript, or in the decision to publish the results.

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
