# Peer review of "Simulation of Groundwater Flow in an Aquiclude for Designing a Drainage System during Urban Construction: A Case Study in Madrid, Spain"

_sustainability, doi:10.3390/su13031526_

Round 1

Reviewer 1 Report

1. In figure 1 include, scale and legends
2. Point 3 (line 131) and point 4 (line 196) have the same subtitle Description of the site and the problem
3. Point 4.1 (line 197), point 4.1.1 (line 198), point 4.1.2 (line 224), 4.4.1 (line 340), 4.4.2 (line 357) present the same subtitle Hydrogeological conceptual model
4. Support with bibliographic reference the statements of:
Line 253
5. Check in equation 2, 120 days does not correspond to 2 months
6. Line 337 correct µS
7. Line 370 correct the number in the figure (figure 7), in figure 7a the colors blue and green are not distinguished. Was the 1. Unify the tense of verbs in the past
2. Write in the first person
3. Improve the quality of all figures
4. Include results tables
5. Separate discussion of results and conclusions
6. In figure 1 include, scale and legends
7. Point 3 (line 131) and point 4 (line 196) have the same subtitle Description of the site and the problem
8. Point 4.1 (line 197), point 4.1.1 (line 198), point 4.1.2 (line 224), 4.4.1 (line 340), 4.4.2 (line 357) have the same subtitle Hydrogeological conceptual model
9. Support with bibliographic reference the statements of:
to. Line 253
10. Check equation 2, 120 days does not correspond to 2 months
11. Line 337 correct µS
12. Line 370 correct the number in the figure (figure 7), in figure 7a the colors blue and green are not distinguished. Were chloride and nitrate ions included in the Piper diagram? Mention its origin in the text
13. Page 13 check superscripts
14. Lines 579-584 mention that the water is of good quality and that there was previously contamination, however, there is no data to prove it
15. Reference point 4.6.2

Author Response

RESPONSES TO THE COMMENT OF THE FIRST REVIEWER

  1. In figure 1 include, scale and legends

It includes graphic scale in both planes. Map legend is not added because there are very few elements to represent, but it is indicated in the figure caption that it is the urbanization of “Los Ahijones”.

  1. Point 3 (line 131) and point 4 (line 196) have the same subtitle Description of the site and the problem

It was a slip of the tongue, point 4 was "Results". Corrected.

  1. Point 4.1 (line 197), point 4.1.1 (line 198), point 4.1.2 (line 224), 4.4.1 (line 340), 4.4.2 (line 357) present the same subtitle Hydrogeological conceptual model

Other unintentional errors, which were not in the original manuscript. The subtitles have been corrected and are as follow:

4.1.1. Definition of hydrostratigraphic units. Permeabilities and transmissivities.

4.1.2. Geometry and limits of the clayey Miocene “Los Migueles” Stream Fault zone.

4.4.1. Approach to the underground flow system before the works in “Los Ahijones”

  1. Support with bibliographic reference the statements of:

Line 253

This is an oral communication from Felix Escolano, who oversaw the construction of the storm-water tank. There is no report on it. We indicate this in the text. Felix Escolano had already been mentioned in "Acknowledgements". There are two previous geotechnical reports made by Euroconsult for this stormwater tank, but they do not report on the hydrogeological behaviour of the fault.

  1. Check in equation 2, 120 days does not correspond to 2 months

It is true, these are 4 months, it is corrected.

  1. Line 337 correct µS

It is corrected here and also in the following lines

  1. Line 370 correct the number in the figure (figure 7), in figure 7a the colors blue and green are not distinguished. Was the 1. Unify the tense of verbs in the past

Figure 7 corrected. And the order number of the following figures and their references in the text Quality colours in figure 7a is improved. The colours of the water types in the Piper map and diagram are also unified. English is corrected.

2.8 Write in the first person

The English is been reviewed by edited by MDPI English editing service

3.9 Improve the quality of all figures

The figures are presented in the best possible quality.

4.10 Include results tables

There are two tables of results (18 O and D analysis) (table 1) and of hydrogeological parameters (table 2) that have already been included.

5. 11 Separate discussion of results and conclusions

Discussion from conclusions are separated. Results are point 4.

  1. In figure 1 include, scale and legends

This question is duplicated and it has already been answered.

  1. Point 3 (line 131) and point 4 (line 196) have the same subtitle Description of the site and the problem

This question is duplicated and it has already been answered.

  1. Point 4.1 (line 197), point 4.1.1 (line 198), point 4.1.2 (line 224), 4.4.1 (line 340), 4.4.2 (line 357) have the same subtitle Hydrogeological conceptual model

This question is duplicated and it has already been answered.

  1. Support with bibliographic reference the statements of:
  2. Line 253

This question is duplicated and it has already been answered.

  1. Check equation 2, 120 days does not correspond to 2 months

This question is duplicated and it has already been answered.

  1. Line 337 correct µS

This question is duplicated and it has already been answered.

  1. Line 370 correct the number in the figure (figure 7), in figure 7a the colors blue and green are not distinguished. Were chloride and nitrate ions included in the Piper diagram? Mention its origin in the text

Plane Colours are improved. The Piper diagram is referred to in the text. Colours of the water types in the Piper map and diagram are also unified. English is corrected.

  1. Page 13 check superscripts

All superscripts in the manuscript have been checked and corrected

  1. Lines 579-584 mention that the water is of good quality and that there was previously contamination, however, there is no data to prove it

Previously there was no permanent flow and thanks to the drainage of the trench a good quality water stream is created in the urbanization of “Los Ahijones”. When the article referred to the poor quality of the stream water, we were referring downstream, outside “Los Ahijones” Urbanisation, where it flows through more gypsummy soils and is naturally contaminated with sulphates. This non-saline water flow would be improved by mixing the water from the stream also downstream. To avoid so many explanations and confusion it was decided to simplify the idea in the article by deleting a sentence in the text of the article.

Thanks you very much for the review,

Joaquin Sanz

Reviewer 2 Report

I realized that the authors in this manuscript had made massive efforts on building a reasonable conceptual model for the research area. Much different information including geological, hydrogeological, and hydrogeochemical data are integrated. And this model is calibrated through the water levels observed at different points located in the interest area. Based on this well-evaluated conceptual model, the authors finally estimated and quantified the influence of the drainage network on the groundwater flow and quality. Constructing a reasonable conceptual model is of great significance in hydrogeological modeling, which is why I think this manuscript should be published. However, prior to publication, I have two main suggestions that should be considered.

  1. To be honest, this is almost the longest time I have ever spent reviewing a manuscript. I recommend asking the native speaker for English proofing. Meanwhile,  I suggest adjusting the article structure. For instance, the geological background introduced in Section 4 should be classified into Section 3, and it should create a new section for the Numerical simulation part in Section 4.6. At last, I strongly recommend to carefully proofread chapter titles and drawings. Using the same title for different content is not the correct choice, such as the title of Section 3 and 4. The clarity and text of all pictures must be well-edited。
  2. Some technical defects reduce the quality of this manuscript. First, the relevant information of the study area is well introduced, but how it is involved in the conceptual model is rarely mentioned. Thus, I suggest adding more information about numerical modeling processes, such as how this model is generalized. Second, it is about the selection of the model parameters. It is necessary to give a sufficient reason why only one permeability parameter is specified for such a large area. To my knowledge, the Lefranc test is applicable for estimating the k above 1e-7 m/s. Note that, the difference between permeability and hydraulic conductivity. Last, if the parameters are not well evaluated, it is recommended to add a description of parameter uncertainty in the discussion part.

Author Response

RESPONSES TO THE COMMENT OF THE SECOND REVIEWER

I realized that the authors in this manuscript had made massive efforts on building a reasonable conceptual model for the research area. Much different information including geological, hydrogeological, and hydrogeochemical data are integrated. And this model is calibrated through the water levels observed at different points located in the interest area. Based on this well-evaluated conceptual model, the authors finally estimated and quantified the influence of the drainage network on the groundwater flow and quality. Constructing a reasonable conceptual model is of great significance in hydrogeological modeling, which is why I think this manuscript should be published. However, prior to publication, I have two main suggestions that should be considered.

To be honest, this is almost the longest time I have ever spent reviewing a manuscript. I recommend asking the native speaker for English proofing.

The English is been reviewed by edited by MDPI English editing service

 Meanwhile, I suggest adjusting the article structure. For instance, the geological background introduced in Section 4 should be classified into Section 3,

In section 3 the geological background is included in the site description, bringing some text from section 4, and adding the rest of the stratigraphy and geological structure

 and it should create a new section for the Numerical simulation part in Section 4.6.

A new section (section 5) is created for numerical simulation

 At last, I strongly recommend to carefully proofread chapter titles and drawings. Using the same title for different content is not the correct choice, such as the title of Section 3 and 4.

we had several mistakes (some slip of the tongue) in the titles of some sections that have been corrected.

 The clarity and text of all pictures must be well-edited。

The quality of all images is improved

Some technical defects reduce the quality of this manuscript. First, the relevant information of the study area is well introduced, but how it is involved in the conceptual model is rarely mentioned. Thus, I suggest adding more information about numerical modeling processes, such as how this model is generalized.

It refers to the reasoning of Condition "Changing Head Defined" (CHD) to define a fixed and known piezometric height, which is followed badly in the text. We have tried to clarify it by including a figure describing where the CHD has been applied.

 Second, it is about the selection of the model parameters. It is necessary to give a sufficient reason why only one permeability parameter is specified for such a large area. To my knowledge, the Lefranc test is applicable for estimating the k above 1e-7 m/s. Note that, the difference between permeability and hydraulic conductivity. Last, if the parameters are not well evaluated, it is recommended to add a description of parameter uncertainty in the discussion part.

Not only have Lefranc tests been carried out but also a pumping test in the alluvial and some other tests using the trial pits in the clays. Alluvial one has served to confirm the orders of magnitude of the values obtained by Lefranc. The tests on trial pits were the same, but they were few in number. In the gypsums, no different tests were carried out to those of Lefranc.

As it is known, in small diameter boreholes for calculating permeability are lower than in pumping tests but given the low permeable clay nature of this aquiclude they are undoubtedly those indicated here.

In addition, the large number that have been made and their wide spatial distribution, make decrease the uncertainty about the validity of the values of hydraulic conductivity. When a lot of data is available, as is the case, we believe that its application is very useful for assessing the overall permeability of a rock mass, which is why it is used in poorly permeable and cohesive soils.

Thanks you very much for the review,

Joaquin Sanz

Reviewer 3 Report

In a first reading the article that is presented seems to be well structured.

In later readings, much more detailed, it shows that the autotres have a deep knowledge of karstic dynamics and the area where the study was carried out.

In this sense, karsty dynamics is one of the most complex processes that can occur in nature, hence its ignorance. on the contrary, the authors carry out a good analysis of this process, of its dynamics, of the factors that prevail in its formation and dynamics. the graphics made are really good and the discussion and conclusions section is correct.

For these reasons I believe that the article will serve and be an effective reference for further research and technical projects.

Author Response

Dear Reviewer,

Thanks you very much for your response and in addition for the review,

Joaquin Sanz

Round 2

Reviewer 1 Report

I agree with the corrections made.  If possible, increase the font size in the images for a better visualization (figures 1, 9 and 11)